# MoTrans: Customized Motion Transfer with Text-Driven Video Diffusion Models

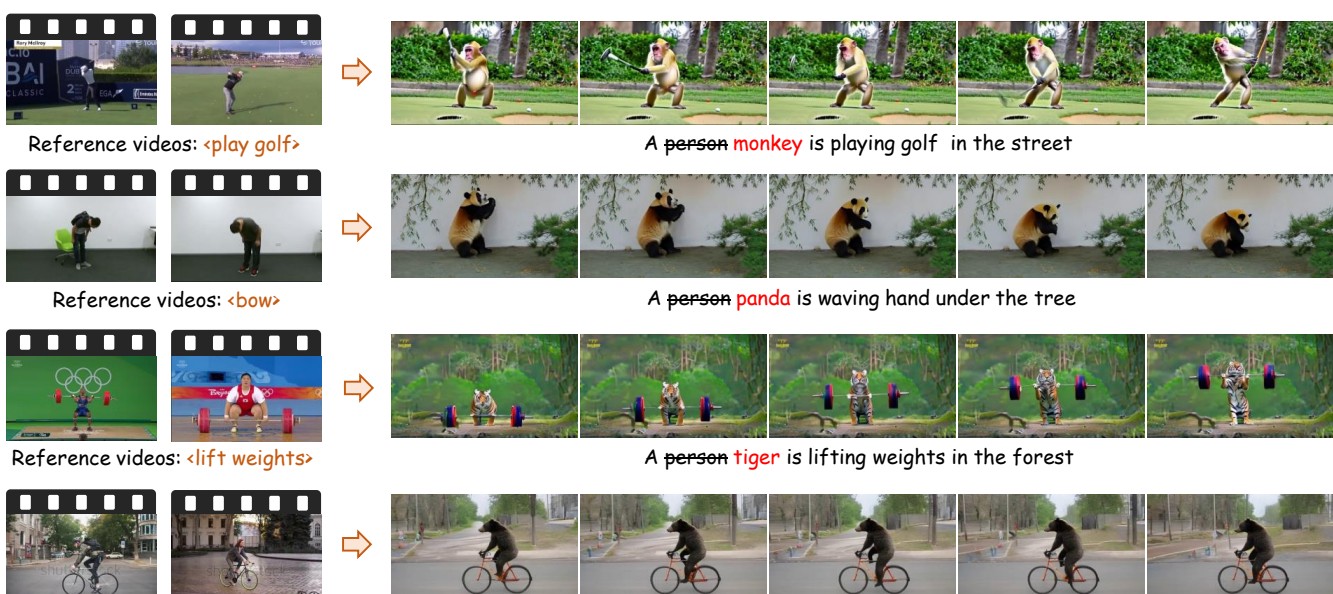

**Figure 1: MoTrans is meticulously crafted to capture precise motion patterns from either singular or multiple reference videos, facilitating seamless transfer of these motions onto fresh subjects within diverse contextual scenes.**

## ABSTRACT

Existing pretrained text-to-video (T2V) models have demonstrated impressive abilities in generating realistic videos with basic motion or camera movement. However, these models exhibit significant limitations when generating intricate, human-centric motions. Current efforts primarily focus on fine-tuning models on a small set of videos containing a specific motion. They often fail to effectively decouple motion and the appearance in the limited reference videos, thereby weakening the modeling capability of motion patterns. To this end, we propose MoTrans, a customized motion transfer method enabling video generation of similar motion in new context. Specifically, we introduce a multimodal large language model (MLLM)-based recaptioner to expand the initial prompt to focus more on appearance and an appearance injection module to adapt appearance prior from video frames to the motion modeling process. These complementary multimodal representations from recaptioned prompt and video frames promote the modeling of appearance and facilitate the decoupling of appearance and motion. In addition, we devise a motion-specific embedding for further enhancing the modeling of the specific motion. Experimental results demonstrate that our method effectively learns specific motion pattern from singular or multiple reference videos, performing favorably against existing methods in customized video generation.

## CCS CONCEPTS

• **Computing methodologies** → **Computer vision**; **Motion capture**.

## KEYWORDS

Diffusion models, Motion customization, Multimodal fusion

## 1 INTRODUCTION

Diffusion-based video generation has achieved significant breakthroughs [3, 11, 17, 33], facilitating the production of high-quality, imaginative videos. While foundation Text-to-Video (T2V) models can generate diverse videos from provided text, tailoring them to generate specific motion could more closely align with user's preferences. Akin to subjects customization in Text-to-Image (T2I) tasks [9, 26, 36], human-centric motions in videos can also be customized and transferred to various subjects, which holds significant practical benefits for animation and film production [45, 46].

ACM MM, 2024, Melbourne, Australia
© 2024 Copyright held by the owner/author(s). Publication rights licensed to ACM.
ACM ISBN 978-x-xxxx-xxxx-x/YY/MM
https://doi.org/10.1145/nnnnnnn.nnnnnnn

Existing pretrained T2V models [5, 42] often struggle to generate intricate, human-centric motions like golf swings and skateboarding, which involve multiple continuous sub-motions. One potential reason is that these foundation models are predominantly trained on highly diverse datasets [1] sourced from the internet, which may suffer from imbalanced data distribution. Consequently, the models might encounter certain motions infrequently, leading to inadequate training for those motions. To better generate particular motions, these pretrained T2V models [5, 42] require fine-tuning on a small set of videos containing the desired motion pattern. However, fine-tuning the model directly without any additional constraints is prone to leading to an undesirable coupling between the motion and the appearance in the limited reference videos and weakening the modeling capability of motion patterns.

Several works [34, 45, 53, 55] have been proposed to address the issue outlined above. These approaches predominantly leverage a dual-branch architecture, with one branch dedicated to capturing single-frame spatial information and the other to inter-frame temporal dynamics. Additionally, they also introduce decoupling mechanisms, such as embedding appearance priors to guide the focus of temporal layers on motion [45] or adjusting latent codes to minimize the negative impact of appearance [55]. Despite their efforts to separate appearance from motion, these approaches exhibit insufficient learning of motion patterns, resulting in videos with diminished motion magnitudes and a deviation from the motion observed in reference videos to some extent.

To this end, we introduce **MoTrans**, a customized **Mo**tion **Trans**fer method, which mainly focuses on modeling the motion patterns in reference videos while avoiding overfitting to its appearance. Specifically, we adopt a two-stage training strategy, with an appearance learning stage and a motion learning stage respectively modeling appearance and motion. To alleviate the coupling issue between appearance and motion, we undertake comprehensive explorations in both stages. 1) During the appearance learning stage, a multimodal large language model (MLLM) is adopted as the recaptioner to expand the original textual descriptions of the reference video. 2) During the motion learning stage, before adapting the temporal module to a specific motion, representations of video frame are pre-injected to compel this module to capture motion dynamics. The complementary multimodal information from expanded prompt and video frame promotes the modeling of appearance and decomposition of appearance and motion. Notably, it has been observed that motions in videos are primarily driven by verbs within the prompt. Inspired by this observation, we employ a residual embedding to enhance the token embeddings of the verbs corresponding to motion, thereby capturing the specified motion patterns in the reference video. Extensive experimental results demonstrate that our method effectively mitigates the issue of overfitting to appearance and produces high-quality motion, performing favorably against other state-of-the-art methods. The main contributions of our work can be summarized as follows:

- We propose MoTrans, a customized video generation method enabling motion pattern transfer from single or multiple reference videos to various subjects.
- By introducing an MLLM-based recaptioner and appearance prior injector, we leverage complementary text and image

multimodal information to model the appearance information, effectively mitigating the issue of coupling between motion and the limited appearance.
- We introduce the motion-specific embedding, which is integrated with temporal modules to collaboratively represent specific motion within reference videos.
- Experimental results demonstrate that our method surpasses other motion customization methods, enabling any motion customization contextualized in different scenes.

## 2 RELATED WORK

### 2.1 Text-to-Video Generation

Diffusion models are catalyzing rapid advancements in image generation tasks [8, 31, 35] and have spawned numerous valuable applications [12, 28, 43, 49, 51, 54]. This success has garnered significant interest in extending these capabilities to video generation [16, 32, 44, 46]. Early efforts in T2V domain [19, 21, 37, 44] primarily focus on cascading video frame interpolation and super-resolution models to generate high-resolution videos, which seems to be complex and cumbersome. In contrast, ModelScopeT2V [42] represents a significant shift by incorporating spatio-temporal blocks atop stable diffusion [35] to model motion more effectively. Building on this, ZeroScope [5] expands the training data and utilizes watermark-free data for fine-tuning, enabling the generation of videos with improved resolution and enhanced quality.

Recently, a new wave of high-quality T2V models [14, 17, 52, 56] has achieved impressive progress. Emu Video [14] generates high-quality videos from natural language descriptions by dividing the video generation process into two steps: initially generating a text-conditioned image, followed by creating videos conditioned on both the text and the generated image. VideoCrafter2 [7] utilizes low-quality videos to ensure motion consistency while employing high-quality images to enhance video quality and conceptual composition ability. Commercial models such as Pika [33] and Gen-2 [11] also exhibit exceptionally strong generative capabilities. Moreover, OpenAI's recent launch of the Sora model [3], capable of generating high-quality videos up to 60 seconds in length, marks a significant milestone in video generation. Although the above foundation T2V models can generate appealing videos, they face challenges in precisely controlling the generated motion.

### 2.2 Customized Video Generation

Existing T2V models [5, 7, 42] excel at generating simple motions or camera movements, struggling to produce specific human-centric motions that align with user preferences. To this end, some models have been introduced to synthesize specific motion pattern and transfer it to diverse subjects. For customized motion transfer [6, 22, 48], some methods employ additional pose maps [4] or dense poses [15] as guidance and require substantial amounts of training data. During the inference stage, it is possible to animate static characters by merely providing initial noise, a reference image, and a set of pose sequences as additional guiding conditions. These approaches allows to produce animations without any need for fine-tuning once they are adequately trained. However, they primarily focus on human-to-human motion transfer and often

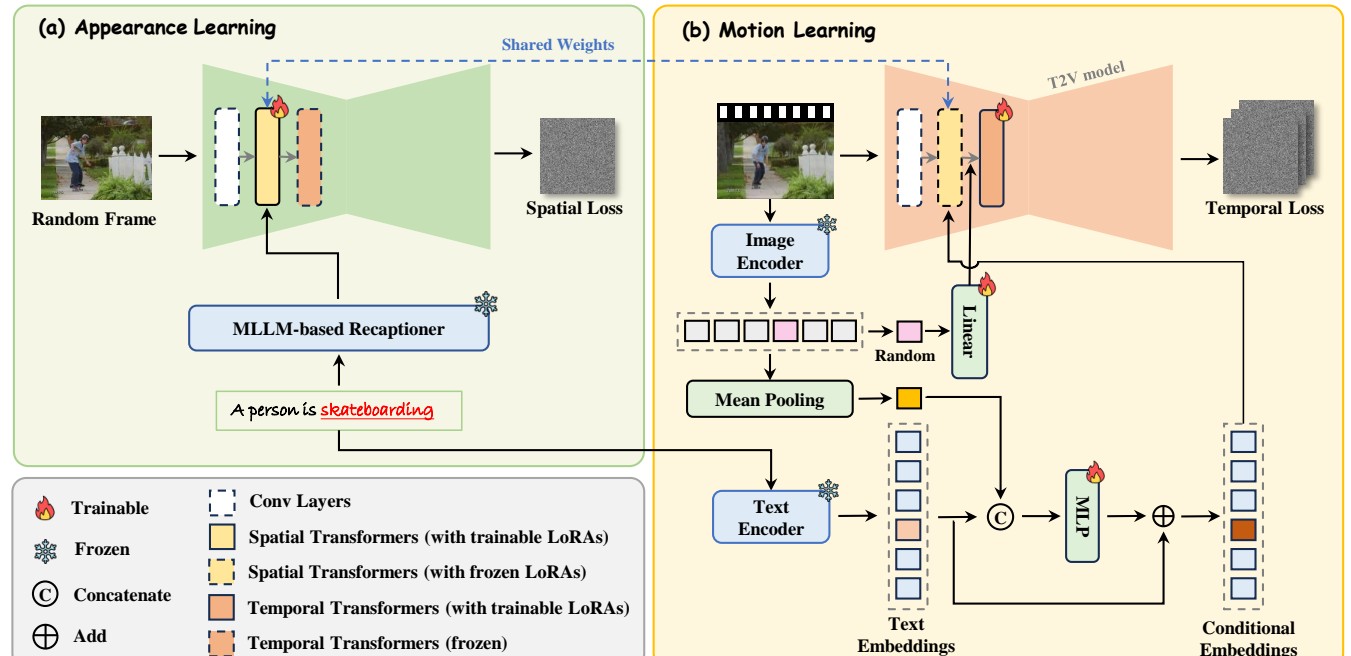

**Figure 2: Overview of the proposed MoTrans. In the appearance learning stage, an MLLM-based recaptioner is employed to extend the base prompt, encouraging the spatial LoRAs to sufficiently learn appearance information. The weights of spatial LoRAs are shared in the second stage. In the motion learning stage, video frame embeddings are injected as appearance priors, compelling the temporal LoRAs to concentrate on motion learning. Furthermore, we adopt MLP to learn a motion-specific embedding, which is jointly trained with the temporal LoRAs to fit specific motion patterns in the reference video.**

struggle to transfer motion to subjects that significantly deviate from the human domain, such as animals.

We aim to learn specific motion patterns rather than precisely replicate every frame's action. This task [29, 34, 45, 47, 53, 55] requires only a minimal amount of training data sharing the same motion concept. Similar to the T2I method DreamBooth [36], these approaches necessitate individual training for each type of motion. Since the generation process does not require additional control conditions such as pose, the resulting motions are more flexible and do not need to follow each frame of the reference video strictly. MotionDirector [55] learns both camera movement and motion, adopting a dual-path way framework to separately learn appearance and motion. During motion learning, the spatial layers trained for appearance learning are frozen to inhibit the temporal layers from learning appearance. DreamVideo [45] introduces structurally simple identity and motion adapters to learn appearance and motion, respectively. To decouple spatial and temporal information, it proposes injecting appearance information into the motion adapter, forcing the temporal layers to learn motion.

Although some methods [34, 45, 53, 55] realize the issue of appearance-motion coupling, they are still prone to synthesizing videos overfitting to the appearances of training data to a certain extent, thereby exhibiting insufficient learning of motion patterns. Furthermore, some methods [47, 53] learn motions that are easier to model. In this paper, we are more concerned with challenging actions with larger ranges of motion, such as sports actions.

## 3 METHOD

### 3.1 Overview

Given a single video or multiple videos with similar motions, the goal of our task is to learn the specific motion or the common motion pattern contained in reference videos. Subsequently, the learned motion can be adapted to new subjects contextualized in different scenes. As illustrated in Fig. 2, the overall training pipeline is divided into the appearance learning stage and the motion learning stage. In the appearance learning stage, we employ an MLLM-based recaptioner to expand the initial prompt of the reference videos. It could promote the modeling capabilities of the spatial attention modules for appearance information. At this stage, we only train spatial low-rank adaptions (LoRAs) and share the weights with the second stage to fit the appearance of the corresponding reference video. As shown in Fig. 4 (a), to preserve the textual alignment capability of the pretrained T2V model, we freeze the parameters of the cross-attention layer and only inject LoRAs into the self-attention and feed-forward layers (FFN). In the motion learning stage, before adapting temporal modules to a specific motion, image embeddings are injected to introduce appearance priors, thereby forcing the temporal LoRAs to focus on motion modeling. Additionally, we employ a multilayer perceptron (MLP) to augment the token embeddings corresponding to verbs, which is jointly trained with temporal LoRAs to capture specific motion pattern. For temporal

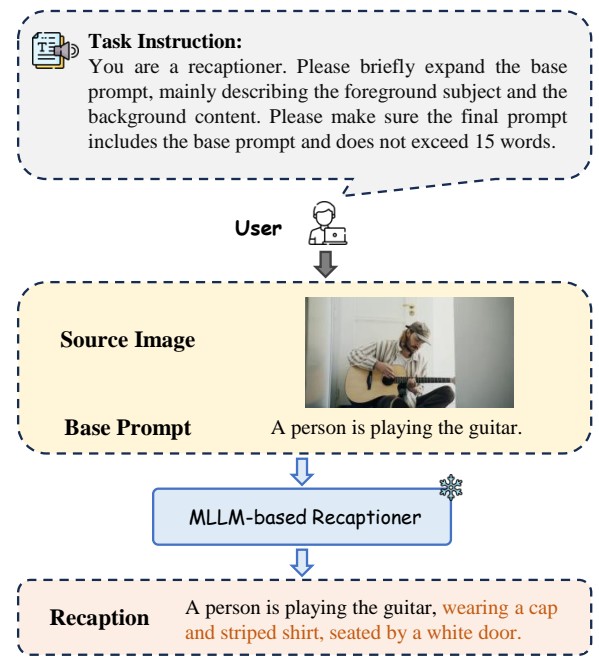

Figure 3: Illustration of multimodal recaptioning. Given an image, an MLLM-based recaptioner is employed to expand the base prompt according to the task instruction, enabling the extended prompt to fully describe its appearance.

modules, LoRAs are injected into both the self-attention layer and FFN of the temporal transformers.

During the inference stage, we integrate the temporal LoRAs and the residual embedding into pretrained video generation models to transfer the specific motion to new subjects.

### 3.2 Multimodal Appearance-Motion Decoupling

The primary objective of our task is to learn motion patterns specified by several reference videos. Due to the inherent characteristics of the diffusion model's training loss, the leakage of some appearance information is inevitable in the motion learning stage. To seperate motion from appearance to a certain extent during the motion learning process, we propose an MLLM-based recaptioner and an appearance injector. In this manner, the complementary multimodal appearance information provided by text and video facilitates the decoupling of appearance and motion information.

**MLLM-based recaptioner.** MLLMs like LLaVA 1.5 [24] or GPT4 [30] have robust in-context reasoning and language understanding capabilities, which can be used for image recaptioning [2, 50]. As illustrated in Fig. 3, let $\mathcal{V} = \{f^i | i = 1, ..., l\}$ denote the reference video with $l$ frames, given a carefully crafted task instruction, the recaptioner can perform text-to-text translation and expand the base prompt $\mathbf{c}_b$ based on a random frame $f^i$. In this manner, the recaptioned prompt $\mathbf{c}_r$ can comprehensively describe the appearance information contained within the video frames. Through training, the spatial attention module will adapt to the appearance information of the reference video and remains frozen in the subsequent

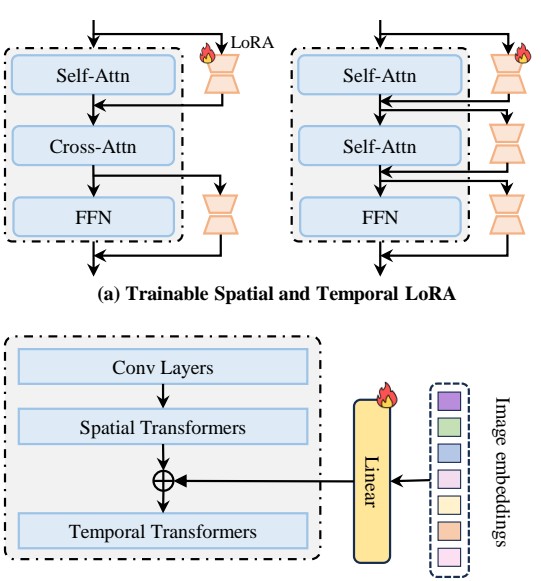

Figure 4: Details of trainable LoRAs and appearance injector. (a) Parameters of the base model are frozen and only parameters of LoRAs injected into the self-attention and FFN are updated. (b) The image embedding is processed through a Linear layer before being fused with the hidden states from the spatial transformers. This pre-injected appearance prior encourages the temporal LoRAs to effectively capture motion patterns.

stage, encouraging the temporal attention module to effectively model the motion information in the videos.

During the appearance learning stage, we adopt a frozen MLLM and only spatial LoRAs need to be trained. The optimization process of this stage is defined as follows:

$$\mathcal{L}_s = \mathbb{E}_{\mathbf{z}_0^i, \mathbf{c}_r, \epsilon \sim \mathcal{N}(0,I), t}[||\epsilon - \epsilon_\theta(\mathbf{z}_t^i, \tau_\theta(\mathbf{c}_r), t)||_2^2]. \quad (1)$$

Here, a VQ-VAE [23] initially compresses frame $f^i$ into a latent representation $\mathbf{z}_0^i \in \mathbb{R}^{b \times 1 \times h \times w \times c}$, where $b, h, w, c$ represent batch size, height, width, and channel count, respectively. $\mathbf{z}_t^i$ is the noised latent code at timestep $t \sim \mathcal{U}(0, T)$. $\tau_\theta(\cdot)$ denotes the pretrained OpenCLIP ViT-H/14 [10] text encoder. Meanwhile, the network $\epsilon_\theta(\cdot)$ is trained to predict the noises added at each timestep.

**Appearance injector.** In addition to leveraging recaptioned prompts to enhance the modeling of appearance, integrating embedding information from video frames themselves can also yield significant benefits. These two modalities collaboratively contribute to the effective decomposition of motion from its appearance. Drawing inspiration from [45], we inject appearance information in the second stage to diminish its impact on motion learning. As shown in Fig. 2 (b), an image encoder $\psi$ is utilized to obtain embeddings of all video frames, we randomly select an image embedding $\psi(\mathbf{f}^i) \in \mathbb{R}^{1 \times d}$ from the input video, where $d$ denotes the dimension of image embedding. Then the appearance information is injected before the temporal transformers, as demonstrated in Fig. 4 (b).

Formally, for each UNet block $l$, the spatial transformer produces the hidden states $h_s^l \in \mathbb{R}^{(b \times h \times w) \times f \times c}$. We employ a linear projection to broadcast the input embeddings across all frames and spatial positions, which are then summed with the hidden states $h_s^l$ before being fed into the temporal transformer. In this way, the appearance representations from the visual modal are pre-injected. The entire process can be formulated as follows:

$$h_t^l = h_s^l \odot (W_p \cdot \psi(\mathbf{f^i})), \tag{2}$$

where $W_p$ represents the weights of the linear projection layer, with its output dimension adapting to variations in the dimensions of the UNet hidden states. And $\odot$ denotes the broadcast operator.

## 3.3 Motion Enhancement

An intuitive observation is that motion patterns in videos generally align with verbs in a text prompt. Hence, we posit that emphasizing verbs could potentially encourage the model to enhance its modeling of motion in the reference videos.

**Motion Enhancer.** A heuristic strategy for enhancing the modeling of motion involves leveraging visual appearance information to enrich the textual embedding representation of motion concept. This is achieved by learning a residual motion-specific embedding on top of the base text embedding. The base embedding can be considered a coarse embedding corresponding to a general motion category, whereas the residual embedding is tailored to capture the specific motion within given reference videos.

Specifically, we employ a pretrained text encoder $\tau_\theta$ to extract text embeddings from a sequence of words $\mathcal{S} = \{s_1, ..., s_N\}$. To locate the position $i$ of the verb $s_i$ in the text prompt, we use Spacy for part-of-speech tagging and syntactic dependency analysis. Following this, the base motion embeddings corresponding to the motion concept are then selected. As shown in Fig. 2 (b), video frames are initially processed by an image encoder to generate frame-wise embeddings. To capture temporal interactions, these embeddings are aggregated through a mean pooling operation, resulting in a unified video embedding. This video embedding is concatenated with the base motion embedding and further processed by an MLP. The MLP comprises two linear layers separated by a Gaussian Error Linear Unit (GELU) [18] activation function. Subsequently, we compute a residual embedding, which is added to the base motion embedding to form an enhanced motion-specific representation. Mathematically, let $E_b$ and $E_r$ represent the base embedding and learnable residual embedding, respectively, the operation can be expressed as follows:

$$E_r = W_2 \cdot (\sigma_{GELU}(W_1 \cdot ([MeanPool(\psi(\mathcal{V})), \tau_\theta(\mathbf{s_i})]))), \tag{3}$$

$$E_{cond} = E_b + E_r. \tag{4}$$

Here, $[\cdot]$ refers to the concatenation operation, and $W_1$ and $W_2$ denote the weights of two Linear layers in MLP. The GELU function is represented by $\sigma_{GELU}$. This motion-specific embedding is integrated with the text embeddings of other words in the prompt to serve as the new condition for training temporal transformers.

To prevent the learned residuals from becoming excessively large, akin to the strategy in [13], we introduce an L2 regularization term as a constraint as:

$$\mathcal{L}_{reg} = ||E_r||_2^2 \tag{5}$$

Similar to the appearance learning stage, the loss function in the motion learning stage calculates the Mean Squared Error (MSE) loss between the predicted noise of the diffusion model and the ground truth noise, except that the frame dimension is no longer 1. Therefore, the final loss function for this stage is defined as:

$$\mathcal{L}_t = \mathbb{E}_{\mathbf{z_0^{1:N}}, \mathbf{c}_b, \epsilon \sim \mathcal{N}(0,I), t} [||\epsilon - \epsilon_\theta(\mathbf{z_t^{1:N}}, \tau_\theta(\mathbf{c}_b), t)||_2^2]. \tag{6}$$

For motion learning, the loss function is the combination of temporal loss and a constraint term as follows,

$$\mathcal{L}_{motion} = \mathcal{L}_t + \lambda \mathcal{L}_{reg}, \tag{7}$$

where $\lambda$ controls the relative weight of the regularization term.

## 4 EXPERIMENTS

### 4.1 Experimental Setup

**Dataset.** We collect a dataset that includes 12 distinct motion patterns, sourced from the Internet, the UCF101 dataset [40], the UCF Sports Action dataset [39], and NTU RGB+D 120 [25]. Each motion pattern is represented by approximately 4-10 training videos. The dataset consists of various sports motions, such as weightlifting and golf swing, alongside large-scale limb movements like waving hands and drinking water. For evaluation, we employ six base prompt templates that involve variations in subject, motion, and context. An example template is "A {cat} is {motion} {in the living room}", with placeholders indicating dynamic elements. Videos corresponding to each motion are generated based on these six prompt categories. More details of our dataset are available in the supplementary materials.

**Implementation details.** We employ ZeroScope as the base T2V model, which is trained with the AdamW [27] optimizer across approximately 600 steps with a learning rate of $5e - 4$. For the spatial and temporal transformers, we specifically fine-tune LoRAs instead of all parameters, with the LoRA rank set to 32. The image encoder used for appearance injection is derived from OpenCLIP ViT-H/14, which is also used to calculate CLIP-based metrics. The regularization loss coefficient for normalizing the verb's residual embedding is $1e - 4$. During inference, we employ DDIM [38] sampler with 30-step sampling and classifier-free guidance scale [20] of 12. We generate 24-frame videos at 8 fps with a resolution of $576 \times 320$. All experiments are conducted on a single NVIDIA A100 GPU.

**Comparison methods.** To investigate the generative capabilities of existing T2V models, we compare our approach with prominent open-source models, including ZeroScope [5] and VideoCrafter2 [7]. Additionally, we explore the effectiveness of directly fine-tuning ZeroScope on a small set of videos containing a specific motion. It is noteworthy that fine-tuning is not applied to the entire diffusion model but specifically targets the LoRAs within the temporal transformers. Our proposed method is adaptable for both single and multiple video customization scenarios. Consequently, we benchmark our approach against open-source methods specialized for one-shot customization, such as Tune-a-Video [46], and for few-shot customization, like LAMP [47]. Further comparisons are conducted with MotionDirector, which serves as our baseline.

**Evaluation metrics.** The performance of the comparison methods is evaluated by four metrics. CLIP Textual Alignment (**CLIP-T**) is employed to assess the correspondence between the synthesized

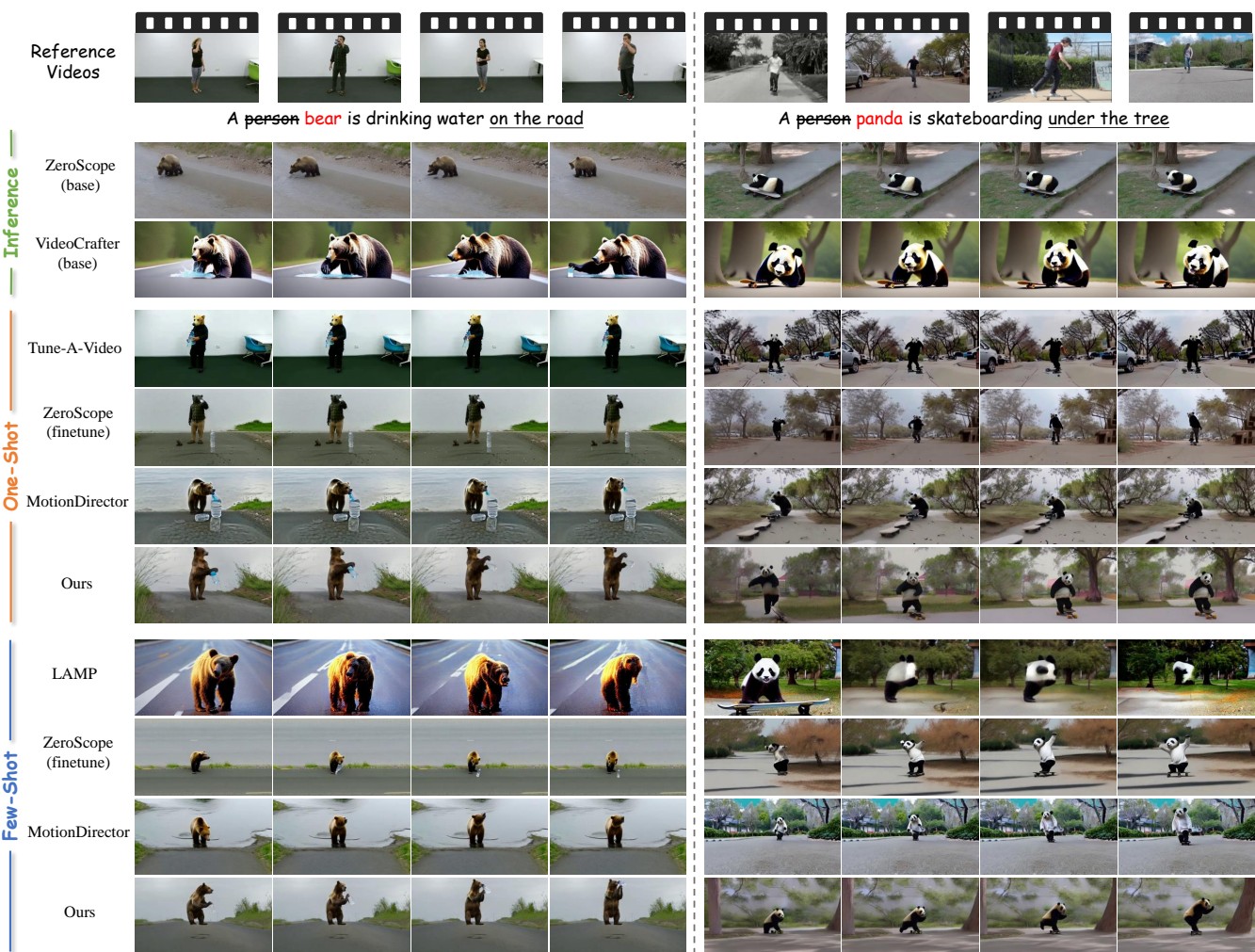

**Figure 5: Qualitative comparison of customized motion transfer. The reference videos on the left demonstrate the motion of a person slowly lifting their hand to drink water. On the right, the videos show a skateboarding pushing action, where the person pushes off the ground with their foot and then slides forward. For one-shot motion customization, the learned motion refers to the second example from the reference videos.** *Best viewed zoomed-in.*

video and the provided prompt, while Temporal Consistency **(TempCons)** measures frame consistency within videos. Due to issues of appearance overfitting observed in some comparison methods, we introduce CLIP Entity Alignment **(CLIP-E)** metric, which is similar to Textual Alignment but foucses on prompts containing only entities, such as "a panda". This metric evaluates whether the synthesized video accurately generates the entity specified by the new prompt. To the best of our knowledge, there exists no metric capable of measuring the congruence between motion patterns in the synthesized videos and those in the reference videos. Therefore, we propose Motion Fidelity **(MoFid)**, which is based on the video understanding model VideoMAE [41]. Specifically, a video $v_m^i$ is randomly selected from the training videos, and a pretrained VideoMAE $f(\cdot)$ is used to obtain the embeddings for both the selected video $v_m^i$ and the synthesized video $\bar{v}_k$. Formally, motion fidelity is

calculated as follows:

$$\mathcal{E}_m = \frac{1}{|\mathcal{M}||\bar{v}_m|} \sum_{m \in \mathcal{M}} \sum_{k=1}^{|\bar{v}_m|} cos(f(v_m^i), \bar{v}_k), \qquad (8)$$

where $\mathcal{M}$ denotes the set of motions, $|\bar{v}_m|$ is the number of videos with motion $m$ in the generated videos, and $cos(\cdot)$ refers to cosine similarity function. Further details on motion fidelity are available in the supplementary material.

## 4.2 Main Results

**Qualitative Evaluation.** To validate the motion customization capabilities of our method, we conduct a comparative analysis with several representative open-source methods tailored for one-shot and few-shot motion customization. As depicted in Fig. 5, direct inference using pretrained T2V models ZeroScope and VideoCrafter

Table 1: Quantitative evaluation of customized motion transfer methods. The best results under one-shot and few-shot settings are highlighted in blue and red, respectively.

| | | CLIP-T (↑) | CLIP-E (↑) | TempCons (↑) | MoFid (↑) |
|---|---|---|---|---|---|
| Inference | ZeroScope [5] | 0.2017 | 0.2116 | 0.9785 | 0.4419 |
| | VideoCrafter [7] | 0.2090 | 0.2228 | 0.9691 | 0.4497 |
| One-shot | Tune-a-video [46] | 0.1911 | 0.2031 | 0.9401 | 0.5627 |
| | ZeroScope (fine-tune) | 0.2088 | 0.2092 | 0.9878 | **0.6011** |
| | MotionDirector [55] | 0.2178 | 0.2130 | **0.9889** | 0.5423 |
| | MoTrans (ours) | **0.2192** | **0.2173** | 0.9872 | 0.5679 |
| Few-shot | LAMP [47] | 0.1773 | 0.1934 | 0.9587 | 0.4522 |
| | ZeroScope (fine-tune) | 0.2191 | 0.2132 | 0.9789 | 0.5409 |
| | MotionDirector | 0.2079 | 0.2137 | 0.9801 | 0.5417 |
| | MoTrans (ours) | **0.2275** | **0.2192** | **0.9895** | **0.5695** |

Table 2: Quantitative results of the ablation study.

| | | CLIP-T (↑) | CLIP-E (↑) | TempCons (↑) | MoFid (↑) |
|---|---|---|---|---|---|
| One-shot | w/o MLLM-based recaptioner | 0.2138 | 0.2101 | 0.9865 | 0.6129 |
| | w/o appearance injector | 0.2114 | 0.2034 | 0.9862 | **0.6150** |
| | w/o motion enhancer | 0.2164 | 0.2135 | 0.9871 | 0.5643 |
| | MoTrans | **0.2192** | **0.2173** | **0.9872** | 0.5679 |
| Few-shot | w/o MLLM-based recaptioner | 0.2179 | 0.2138 | 0.9792 | 0.5997 |
| | w/o appearance injector | 0.2143 | 0.2132 | 0.9807 | **0.6030** |
| | w/o motion enhancer | 0.2211 | 0.2171 | 0.9801 | 0.5541 |
| | MoTrans | **0.2275** | **0.2192** | **0.9895** | 0.5695 |

fails to synthesize specific motion patterns due to the lack of fine-tuning on specified videos. Additionally, the synthesized videos exhibit notably small motion amplitudes, suggesting that pretrained T2V models struggle to generate complex, human-centric motion. In particular, these models face significant challenges in generating specific motions without targeted training on specific videos. Furthermore, unconstrained fine-tuning of Zeroscope leads to an undesirable coupling between appearance and motion, and the motions in the generated videos do not sufficiently resemble those in the reference videos, with notably small motion amplitudes.

Tune-A-Video, which targets single-video customization and is based on the T2I model, suffers from poor inter-frame smoothness and severe appearance overfitting. Similarly, the few-shot motion customization method LAMP, also leveraging a T2I model, exhibits very poor temporal consistency and heavily relies on the quality of the initial frame. Compared to other methods, LAMP requires more reference videos and training iterations to achieve relatively better results. MotionDirector also encounters challenges with appearance overfitting, often generating unrealistic scenarios such as a panda on a skateboard dressed in human attire. Moreover, it exhibits insufficient modeling of motion patterns, resulting in videos with diminished motion magnitudes and deviations from the observed motion in reference videos.

Our method, however, demonstrates superior ability to accurately capture motion patterns in both one-shot and few-shot motion customization scenarios. Additionally, one-shot methods sometimes fail to discern whether to learn camera movement or foreground motion. In contrast, few-shot methods can leverage inductive biases derived from multiple videos, better capturing common motion patterns. This allows the temporal transformer to focus on foreground action rather than camera movements.

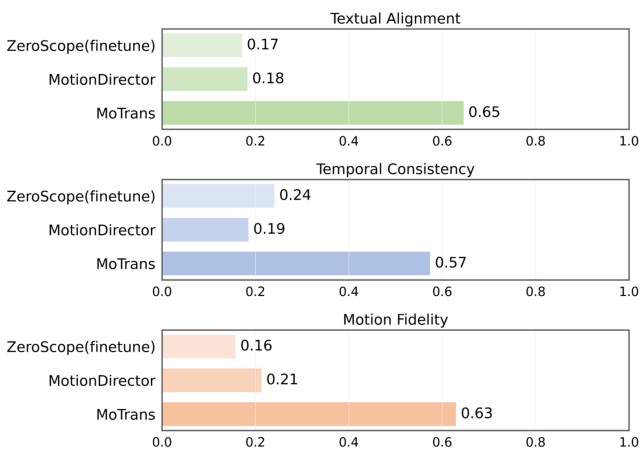

Figure 6: User study. For each metric, the percentages attributed to all methods sum to 1. MoTrans accounts for the largest proportion, indicating that the videos generated by our method exhibit superior text alignment, temporal consistency, and the closest resemblance to the reference video.

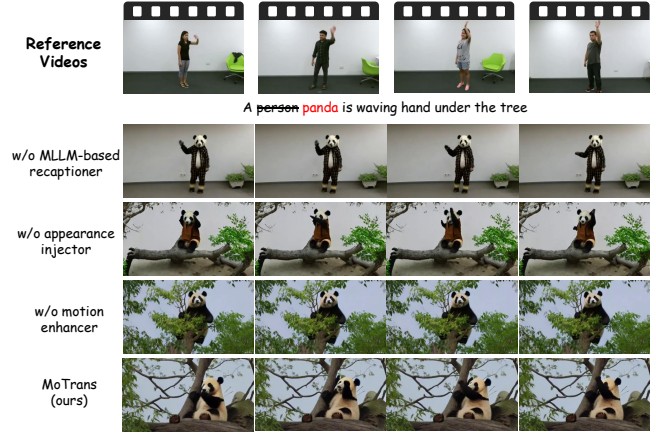

Figure 7: Qualitative results of the ablation study. Given several reference videos, Motrans can learn motion patterns from reference videos without appearance overfitting.

**Quantitative Evaluation.** As illustrated in Table 1, when only a single reference video is provided, both Tune-a-Video and the fine-tuned Zeroscope exhibit higher motion fidelity but lower entity alignment. This is primarily attributed to the severe appearance overfitting, which leads to pronounced similarities in both appearance and motion to the reference video. Consequently, these methods fail to synthesize the new subject specified in the prompt, which is also demonstrated in Fig. 5. When multiple reference videos are provided, our approach outperforms other methods across all evaluated metrics. Notably, it achieves high levels of text alignment and motion fidelity, showcasing our method's capability to effectively learn motion patterns from reference videos while avoiding overfitting to the appearance information.

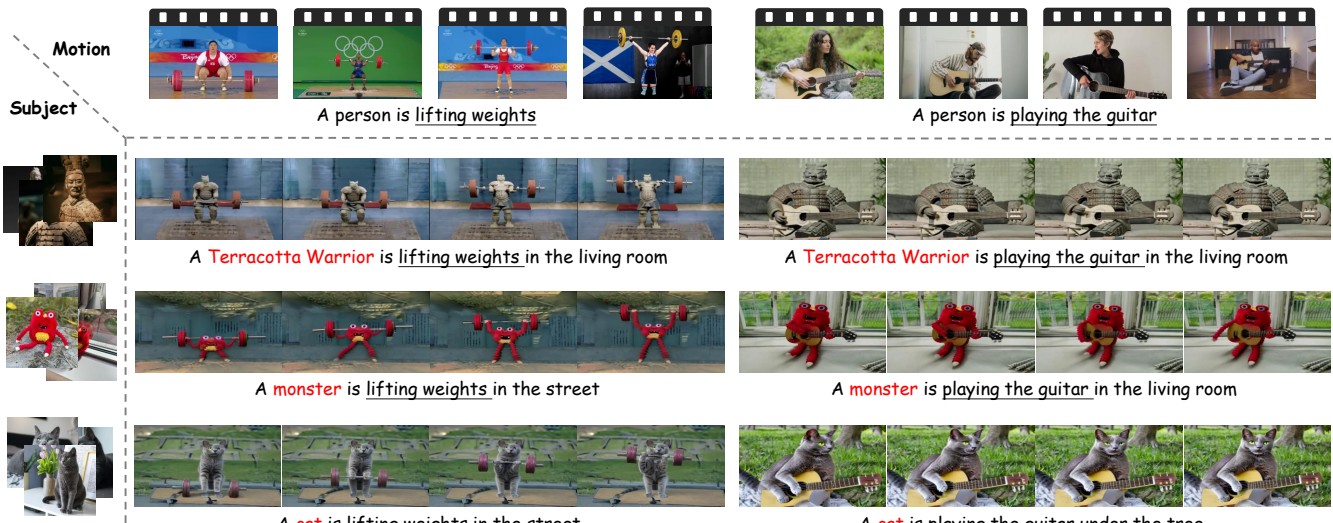

**Figure 8: Customized video generation with specific subjects and motions. The two-stage training strategy allows for the motion transfer (top) from the reference video to the subject specified by exemplar images (left).**

**User study.** Automatic metrics like CLIP-T have limitations in fully reflecting human preferences, hence, we conduct user studies to further validate our method. We collect 1536 sets of answers from 32 participants, with each completing a questionnaire containing 48 sets of questions. Participants are asked to pick the best video through answering the following questions: (1) which video better aligns with the textual description? (2) which video is smoother, and with fewer flickering? (3) which video's motion is more similar to that in the reference video without resembling its appearance? Considering the significantly inferior performance of the T2I-based model LAMP, our comparison primarily focused on MoTrans versus the other methods. The results, shown in Fig. 6, reveal that our method consistently outperforms the others across all metrics, aligning more closely with human intuition.

### 4.3 Ablation Study

We conduct ablation studies to demonstrate the efficacy of the key modules introduced in this paper. Specifically, the MLLM-based recaptioner and the appearance injector leverage the prior knowledge of multimodal sources, i.e., textual and visual modalities, to address the challenge of coupling between appearance and motion. As illustrated in Table 2 and Fig. 7 (rows 1 and 2), the absence of either the MLLM-based recaptioner or the appearance injector leads to a performance drop in both CLIP-T and CLIP-E, alongside high motion fidelity. This suggests a severe overfitting to both appearance and motion. Additionally, without the motion enhancer, the model struggles to synthesize the specific motion depicted in the reference videos, but with the introduction of the first two modules, it can synthesize the specified subject. In comparison, our method effectively mitigates appearance overfitting while ensuring motion fidelity as much as possible. Each module we propose significantly contributes to the improvement of the final generation results.

### 4.4 Application

**Video customization with both subject and motion.** Benefitting from the two-stage training strategy of our approach, appearance and motion can be learned separately through the spatial and temporal transformers within a UNet. As depicted in Fig. 8, we simultaneously customize the subject depicted in an image set and the motion specified by a video set. The customization results demonstrate that our method does not suffer from appearance overfitting to the training data and can successfully enable a specific animal or inanimate object to perform a human-centric motion.

### 5 CONCLUSION

We propose MoTrans, a customized motion transfer method that effectively transfers a specific motion pattern from reference videos to diverse subjects. By integrating multimodal appearance priors, encompassing both visual and textual modalities, our approach mitigates the issue of coupling between motion patterns in synthesized videos and the limited appearance contained in reference videos. Additionally, our method employs dedicated residual embeddings to accurately represent the specific motion pattern inherent in the reference videos. Compared with existing methods, our method demonstrates superior capabilities in customizing motion and decoupling appearance, and it also supports the simultaneous customization of subjects and motions.

Although our method can synthesize high-quality motion, it is currently optimized for short video clips of 2-3 seconds and faces challenges in generating longer sequences. Moreover, while our method currently supports the customization of motion for a single subject, extending this capability to multiple subjects performing the same motion remains a challenge. Future work will aim to address these limitations and expand the applicability of our method to more complex and practical scenarios.

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
