# OpenReview forum: "MoTrans: Customized Motion Transfer with Text-driven Video Diffusion Models"
_acmmm.org/ACMMM/2024/Conference — MM2024 Poster_

### Official Review · Reviewer_mLBN · 2024-05-19

**Rating:** 4
**Confidence:** 2

**Summary:**

In this paper, the authors propose a customized motion transfer method, MoTrans, to effectively decouple appearance and motion. MoTrans is divided into two learning stages: the appearance learning phase and the motion learning phase. In the appearance learning stage, the Multimodal Large Language Model (MLLM) is employed as the recaptioner to extend the original textual descriptions of the reference video; in the motion learning stage, the temporal module is used to capture motion information from the reference video. Experimental results demonstrate that MoTrans can effectively learn specific motion patterns from singular or multiple reference videos.

**Strengths:**

1. This paper is structured and easy to understand.
2. In the motor learning stage, the temporal module is injected with appearance information to allow the temporal module to learn effective motor information.
3. The paper is complete with comparative experiments to demonstrate its effectiveness by comparing a number of different methods.

**Limitations:**

1. When there is a large change in the background, does the motion learning stage learn background motion features that affect the motion of the foreground?

2. In the appearance learning stage, Spatial Transformers receive prompts that have gone through the MLLM-based recaptioner. But in motion learning stage, why prompt does not need to go through the MLLM-based recaptioner? Please explain it.

3. When generating videos on specific subjects. Is it also possible to achieve good results when there is a large difference in the foreground size of the reference video?

**Suitability:**

3

---

### Official Review · Reviewer_dPjX · 2024-05-19

**Rating:** 5
**Confidence:** 3

**Summary:**

The paper proposes a motion transfer method for video generation which intends to decouple the appearance and motion learning. In the appearance learning branch, the method proposes to use a MLLM-based recaptioner to extend the initial prompts to comprehensive descriptions on the appearance to learn spatial LoRAs. In addition, the method learns a linear layer from the randomly selected image embeddings extracted from video frames as an appearance injector in learning the temporal patterns. In the motion learn branch, the method proposes to use a pretrained text encoders to focus on \emph{verbs} in the text prompt. The motion-specific embedding is combined with the text embeddings to control the video generation within the off-the-shelf ZeroScope T2V model. The experiments show that the proposed method can effectively transfer human motions to animals, such as bear, panda and cat.

**Strengths:**

The proposed motion transfer method is well motivated to extend motion transfer beyond human for general video generation.

The particular way to decouple appearance and motion, such as the recpationer and spatial LoRA as well as the appearance injector, is reasonable and somewhat novel.

Overall, the paper is well written and sufficient experiments.

**Limitations:**

The motion learning branch employs a simple way to learn motion-specific embeddings, e.g., a mean pooling of the embeddings of verbs over frames, which needs more justification how this scheme can learn motion-specific embeddings.

Why a randomly select image embedding from the input video can diminish the impact of appearance to motion learning when being infected before the temporal transformers? Any better scheme to select the appearance injectors?

Please discuss the limitation of the proposed method transferring motion to different kinds of objects, e.g., 4 limbs required to transfer human motion?

Overall, the paper studied a critical problem in controllable video generation and push a step forward, though there is some gap between the proposed approach and the claim that needs to be elaborated.  Nevertheless, the work deserves to share with the community.

**Suitability:**

2

---

### Official Review · Reviewer_tL5r · 2024-05-24

**Rating:** 3
**Confidence:** 4

**Summary:**

Given reference videos and a text prompt, the research aims at customized motion generation. This paper proposed a two-stage method to decouple motion and appearance. In the appearance learning stage, they proposed an MLLM-based recaptioner and an appearance injector. In the motion learning stage, they proposed a residual embedding to enhance the effect of verbs. Finally, the experiments on the collected dataset show the performance of the proposed method.

**Strengths:**

1. This paper explored the customized motion generation under decoupling motion and appearance.

2. The idea to use LLMs to generate the more detailed caption of the source image, is useful to improve the ability of learning appearance.

3. The idea of attention to verbs of the text is also useful in text learning.

4. The paper is easy to follow.

**Limitations:**

1. Lack of details. Though the paper introduces the two stages of the proposed turn, the details of each stage are still vague. For example, In L446, do you mean the method is a latent diffusion based method? Does equation (1) refer to spatial loss? What is spatial LoRA and temporal LoRA? There are no specific definitions for them.

2. Lack of novelty. The proposed method is similar to MotionDirector. For example, the spatial/temporal LoRAs and spatial/temporal loss are the same with MotionDirector. The difference is only the injection of text-image multi-modal information and video information. Hence, the novelty of the proposed method is limited. It can also be proved by the experimental results, which show little improvement compared with MotionDirector.

3. Correctness of the proposed MoFid metric. As mentioned in L630, MoFid measures the motion patterns. But in Table 1, when the proposed method is low on MoFid, they argue the reasons from the appearance. In Table 2, when w/o appearance injector yields a high score on MoFid, they argue without appearance information they got more consistent motion patterns. A correct metric for evaluating the motion pattern should be robust, and not be so influenced by other factors.

4. The experimental results are on the collected dataset. Could you give the results on one public dataset, not a mixed dataset?

**Suitability:**

3

---

### Official Review · Reviewer_2pts · 2024-05-27

**Rating:** 4
**Confidence:** 4

**Summary:**

This paper focused on the text-driven video diffusion task and proposed a customized motion transfer method named MoTrans, which enables video generation of similar motion in new context.  A multimodal large language model (MLLM)-based recaptioner in applied to expand the initial prompt to focus more on appearance and an appearance injection module is introduced to adapt appearance prior from video frames to the motion modeling process. A motion-specific embedding is proposed to further enhance the modeling of the specific motion. Experimental results on UCF101, UCF Sports Action, and NTU RGB+D 120 datasets validate the effectiveness of the MoTrans in learning specific motion pattern from singular or multiple reference videos.

**Strengths:**

+) Clear motivations and contributions.

+) Good writing and the technical details are easy-to-follow.

+) The promising video generation results.

**Limitations:**

-) What about using the reference videos that contain only one frame? In this setting, the temporal coherece is missing and the apperance information dominates. This experiment is important for comprehensvely evaludate the ability of apperance modeling of the MoTrans.

-) Some results in Table 1-2 are not the best, for example, the TempCons and MoFid, more insights behind this phenomenon are needed.

-) Since all the examples contain single object, the videos containing multiple objects are required to be given.

-) What is the computational cost (e.g., several seconds or minutes) for generating one video given the reference video?

**Suitability:**

3

---

### Meta-Review · Area_Chair_7QaY · 2024-07-01

**Recommendation:** Accept (Poster)
**Confidence:** 3

**Metareview:**

The paper introduces MoTrans, a customized motion transfer method designed for text-driven video diffusion, particularly focusing on capturing and replicating specific motion patterns in new contexts. This method employs a multimodal large language model (MLLM)-based recaptioner to enhance initial prompts with a focus on appearance, and introduces an appearance injection module to incorporate appearance priors from video frames into the motion modeling process.

Strengths:

+ MoTrans effectively addresses the text-to-video generation challenge, leveraging both appearance and motion data to enhance video output.

+ The paper is well-structured and clearly written, making technical details accessible and easy to follow.

+ Experimental validation on datasets like UCF101 and NTU RGB+D 120 demonstrates the method's effectiveness in learning specific motion patterns from reference videos.

Limitations:

+ The method's ability to handle scenarios with minimal temporal data, such as reference videos containing only one frame, is not explored, raising questions about its versatility in appearance modeling.

+ Some metrics such as TempCons and MoFid in the experimental results do not outperform existing methods, and the reasons behind these outcomes are not thoroughly explained.

+ The examples provided in the study are limited to scenarios containing a single object, which may not adequately demonstrate the method's effectiveness in more complex, multi-object settings.

+ The computational cost of generating videos is not detailed, which is crucial for evaluating the practical applicability of MoTrans in real-world scenarios.

Conclusion:

MoTrans introduces innovative techniques for motion and appearance integration in video generation, showing promising results. However, further exploration into its limitations and broader applicability is necessary to fully ascertain its potential. The AC suggests a borderline accept, indicating that while it presents significant advancements, additional clarifications and expansions on its capabilities and performance would strengthen its contribution to the field. The authors might want to carefully address the concerns of Reviewer tL5r.